# Systemic Retinoids for Generalized Verrucosis Due to Congenital Immunodeficiency: Case Reports and Review of the Literature

**DOI:** 10.3390/genes14030769

**Published:** 2023-03-22

**Authors:** Hideyuki Kosumi, Ken Natsuga, Teruki Yanagi, Hideyuki Ujiie

**Affiliations:** Department of Dermatology, Faculty of Medicine and Graduate School of Medicine, Hokkaido University, Sapporo 060-8638, Japan

**Keywords:** generalized verrucosis, human papilloma virus, congenital immunodeficiency, retinoids

## Abstract

Generalized verrucosis (GV) is a group of immunodeficiency disorders accompanied by widespread human papillomavirus infection. We revisit two cases of GV due to congenital interleukin-7 deficiency successfully treated with systemic retinoids. We also present a review of the literature on the use of systemic retinoids to treat GV. Our review suggests that systemic retinoids are a safe and effective option for managing recalcitrant wart lesions in cases of GV.

## 1. Introduction

Human papillomavirus (HPV) is a double-stranded deoxyribonucleic acid (DNA) virus belonging to the Papillomavirus genus of the Papillomaviridae family [1,2]. Different HPV types are associated with different diseases and tend to infect specific body sites [3]. Certain HPV types preferentially target cutaneous epithelium and are found in skin lesions such as plantar warts, common warts, and flat warts. Specific HPV types are correlated with distinct skin lesions; for example, α-HPV, such as HPV-2, 27, 57 in common warts; μ-HPV, such as HPV-1 in myrmecia warts; and γ-HPV, such as HPV-4, 60, 65 in pigmented viral warts are well-known pathogens [3]. Cell-mediated immunity is crucial against HPV [4,5], and various immunodeficiencies can result in widespread and persistent wart lesions. This condition is referred to as generalized verrucosis (GV) [6].

GV is defined as disseminated skin HPV infection with more than 20 lesions distributed in more than one localized region of the body [6]. Some GV patients present with characteristic wart lesions induced by genus β HPV (β-HPV), and they are known as epidermodysplasia verruciformis (EV) [7,8] or acquired EV (AEV) [9,10]. EV is a rare genodermatosis characterized by congenital susceptibility to β-HPV due to biallelic inactivation mutations in *TMC6* and *TMC8* [8,11]. The characteristic skin lesions of EV, such as verruca plana-like flat-topped papules on the extremities and pityriasis versicolor-like macules on the trunk and face, correlate with β-HPV including HPV-3, 5, 8, 9, 10, 12, 14, 15, 17, 19–25, 28, 29, 36, 46, 47, 49, and 50 [7,12]. These subtypes are known as EV-HPV because β-HPV serotypes are ubiquitous and nonpathogenic in the normal population [13]. AEV, also known as secondary EV, is a condition with clinical similarities to EV that can occur in patients with acquired cell-mediated immunodeficiencies [9]. AEV can be caused by human immunodeficiency virus (HIV) infection [14,15], non-Hodgkin lymphoma [16], lepromatous leprosy [17], lipoid proteinosis [18], or immunosuppressants [19,20]. 

Although various therapies have been reported, GV tends to be intractable and refractory, especially in cases caused by congenital immunodeficiencies, due to the lack of effective treatments for the underlying immunodeficiencies [6,21]. The main purpose of therapies for GV due to congenital immunodeficiency is therefore to alleviate symptoms such as pain or motor impairment and address cosmetic issues.

We previously reported two cases of GV due to interleukin (IL)-7 deficiency [22,23]. Here we revisit the efficacy of systemic retinoids in these cases and review previous publications, which when taken together, suggest that systemic retinoids may be effective for GV due to congenital immunodeficiency.

## 2. Case Presentation

Cases 1 and 2 are patients we previously reported as cases of GV due to IL-7 deficiency [22,23]. Both patients were homozygous for c.3G > A (p.Met1?) in *IL7* (NM_000880.4) [23]. The treatment course for Case 1 has not been previously described. In brief, Case 1 was a 51-year-old Japanese female who presented with chronic pain due to fissures on the numerous verrucous plaques on the limbs that had appeared 40 years before her referral. She had been treated with the topical application of salicylic acid, cryotherapy, and surgical removal, which displayed limited efficacy. Physical examination revealed papillomatous papules, nodules, and plaques on the whole body. Hyperkeratotic lesions were prominent on the palms and soles (Figure 1a), and oozing from the fissures was observed (Figure 1a, arrow). We treated her with 10 mg/day of etretinate. After four weeks, approximately 80% of the hyperkeratotic lesions and fissures were remarkably reduced (Figure 1b), leading to resolution of the pain, which improved her quality of life.

Case 2 was a 63-year-old Japanese male with GV (Figure 1c). Detailed clinical information was reported previously [22]. As reported, we treated him with 50 mg/day of etretinate and topical maxacalcitol. The treatment dramatically improved the lesions and flattened around 90% of them (Figure 1d). Although the tapering of the etretinate to 30 mg/day caused the recurrence of the lesions, motor impairment and the pain during physical activities were markedly reduced to the extent that the lesions were controlled [22]. We performed serologic testing on the patients one month after the initiation of the treatment and repeated it every two to three months during the treatment. The serological tests monitored liver function, kidney function and triglyceride metabolism, which can be affected by retinoids. There were no serious complications in either case.

## 3. Review of the Literature

We reviewed the literature on GV treated with systemic retinoids (Table 1 and Appendix A), which identified 20 cases, 10 males and 10 females aged between 12 and 87 years old including the resent patients, who had been treated with systemic retinoids [10,21,24,25,26,27,28,29,30]. Acitretin was used to treat 14 out of the 20 patients, etretinate to treat 5, and isotretinoin to treat 1 patient. The efficacy of the systemic retinoids was evaluated in 13 cases and showed moderate to great improvement in all cases except for a patient who could not continue due to intolerance [21]. Of the 12 remaining cases, 3 were treated with a combination of topical therapies such as imiquimod, trichloroacetic acid and maxacalcitol, and the other 9 cases were treated with systemic retinoids alone. Of the 12 cases, 11 (91.7%) showed treatment efficacy. Cessation of the systemic retinoids tended to result in the recurrence of verrucous lesions, but it is worth noting that the one case treated with systemic isotretinoin alone achieved complete remission without recurrence for three years after treatment cessation [28]. No serious complications occurred during treatment in the reported cases.

## 4. Discussion

We reported two cases of GV due to IL-7 deficiency whose lesions and pain were effectively controlled with systemic retinoids. Although various treatments have been proposed, the treatment of GV due to congenital immunodeficiency has not been standardized [6]. Common approaches to cutaneous warts can be classified as (i) chemical or physical ablation of the affected tissue by utilizing methods such as salicylic acid, cryotherapy, cantharidin, trichloroacetic acid, surgery, and laser, (ii) enhancement of the local immune response with imiquimod, topical or intralesional immunotherapy, etc., and (iii) antitumor therapy including topical fluorouracil, bleomycin, and topical or systemic retinoids [6,31]. This article focuses on systemic retinoids as a therapy for GV due to congenital immunodeficiency for the following reasons: (I) topical treatments, including destructive therapies, are not desirable due to the widespread distribution of lesions and the pain associated with the destructive methods used in cases of GV [24,32]; (II) therapies dependent on immune systems such as imiquimod can be less effective due to defects in the anti-HPV immune systems of patients with GV due to congenital immunodeficiency [25]; and (III) systemic retinoids have been reported to be effective in several cases [26]. Although the mechanism how retinoids work in EV treatment is not fully elucidated, retinoids have been reported to reduce the HPV proliferation and HPV-induced epithelial hyperplasia, probably due to effects on keratinocytes, antiviral activity, or killer T-cell increase in the lesions [33]. Our review suggests that (i) systemic retinoids are effective in reducing verrucous lesions and can improve secondary symptoms such as pain, motor impairments, and cosmetic issues; (ii) systemic retinoids are a relatively safe treatment option for GV; and (iii) treatment cessation results in verrucae recurrence in the majority of cases. 

## 5. Conclusions

Based on the efficacy and safety, we believe that systemic retinoids should be considered as a treatment option for GV.

## Figures and Tables

**Figure 1 genes-14-00769-f001:** (**a**,**b**) Clinical photos of Case 1. (**a**) Numerous verrucae were observed on the limbs. Arrow indicates oozing from the fissure. (**b**) One year after treatment with etretinate. (**c**,**d**) Clinical photos of Case 2. (**c**) Verrucous plaques were found on the extremities. (**d**) Four months after treatment with etretinate.

**Table 1 genes-14-00769-t001:** Summary of all published work on GV treated with systemic retinoids.

Pt	Diagnosis	Age	Sex	Treatment	Dose	Course	Ref
1	GV due to IL7 deficiency	51	F	etretinate	10 mg/day	Moderate improvement	[23] (Case 1)
2	GV due to IL7 deficiency	63	M	etretinate	50 mg/day	Moderate improvement	[22] (Case 2)
3	EV	87	F	acitretin	Not mentioned	Not mentioned	[10]
4	EV	60	F	acitretin
5	GV	64	M	acitretin
6	GV	25	M	acitretin
7	GV	87	M	acitretin
8	GV	67	F	acitretin
9	GV due to systemic steroid	54	M	acitretin
10	GV due to GATA2 deficiency	24	M	acitretin	25 mg/day	Could not continue the treatment	[21]
11	GV due to GATA2 deficiency	12	M	acitretin	10 mg/day	Complete improvement after bone marrow transplantation	[24]
12	EV	20	M	isotretinoin	0.8 mg/kg/day	Complete remission	[26]
13	EV	25	F	acitretin	0.5–1 mg/kg/day	Moderate improvement	[27]
14	GV	16–39	M	Etretinate for three, acitretin for two	Not mentioned	Good results with rapid disappearance (<1 month) One had complete remission without recurrence after cessation	[28]
15	GV	F
16	GV	F
17	GV	F
18	GV due to HIV	F
19	GV	43	M	acitretin	0.7 mg/kg/day	Cosmetic improvement	[29]
20	Acquired EV due to HIV infection	46	M	acitretin	Not mentioned	Successfully maintained with topical imiquimod	[30]

## Data Availability

Not applicable.

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
