# Peer review of "Systemic Retinoids for Generalized Verrucosis Due to Congenital Immunodeficiency: Case Reports and Review of the Literature"

_genes, 2023, doi:10.3390/genes14030769_

Round 1
Reviewer 1 Report
Regarding 2 cases with GV,
1 Did the authors did more investigations besides clinical diagnosis; for example, histopathology, RT-PCR for identifying types of HPV.
2 Why did you choose etretinate not acitretin in these cases? Are there any reports about different response of HPV to each drug?
3 Did you observed side effects during treatment period?
Author Response
Response to the reviewers
RE: genes-2198240 revised version R1
Systemic retinoids for generalized verrucosis due to congenital immunodeficiency: case reports and review of the literature
Response to Reviewer 1
Comment:
Regarding 2 cases with GV,
1 Did the authors did more investigations besides clinical diagnosis; for example, histopathology, RT-PCR for identifying types of HPV.
We thank the reviewer for the valuable suggestions for improving the manuscript. In the case 1, histopathologic examination showed acanthosis with koilocytosis. Three wart lesions were tested for HPV typing with a PCR method, which detected HPV 7 in all of the lesions. In the case 2, a histology sampled from a hyperkeratotic erythematous plaque on the right forearm revealed an acanthotic epidermal layer with sharply defined nests of basaloid cells. The nonhyperkeratotic pigmented macule on his upper back showed papillomatosis, hyperkeratosis and vacuolization of the cytoplasm of granular cells with coarse keratohyalin granules. PCR for HPV typing with skin lesions detected HPV 3, 50, 57 and 76. All of the above is precisely reported in the previous reports from our group (Kosumi H et al, Clin Infect Dis, 2020, and Yanagi T et al, Clin Exp Dermatol, 2006.) and we have omitted the information to avoid self-plagiarism.
2 Why did you choose etretinate not acitretin in these cases? Are there any reports about different response of HPV to each drug?
There are no reports of different responses for warts or generalized verrucosis with each retinoid. We chose etretinate because both acitretin and isotretinoin are not available in Japan.
3 Did you observed side effects during treatment period?
There were no serious complications in either case. We have included this information in the revised manuscript.

Reviewer 2 Report
1 Good attempt by the authors, however the study lacks novelty as systemic retinoids are a known option for the management of warts especially in the setting of immunodeficiency.
2 Nicely written article, however focus of the manuscript should be on mechanism of action of retinoids in warts in general, in immunodeficiency disorders like GV, EDV and any specific proposed mechanism of action in IL-7 deficiency cases as in the present 2 cases.
3 It would be good to mention the percentage improvement noted by care giver in the lesions. Also, mention the patient satisfaction with treatment, especially if it improved their quality of life.
4 In table 1, mention the duration of therapy as well.
5 Retinoids are given not only for the improvement of warts, but they also have roles as immunomodulators and prevention of skin cancers in conditions like EDV. Authors should mention this as well along with appropriate references.
6 Since long term retinoids are generally needed for the management of warts in GV, a note on how to monitor therapy (like what investigations to be done and how frequently) and side effects of long term therapy should be mentioned for the benefit of readers.
7 Mention the success rate of improvement with retinoids for treatment of warts in general and in immunodeficiency disorders based on the available literature.
Author Response
Response to the reviewers
RE: genes-2198240 revised version R1
Systemic retinoids for generalized verrucosis due to congenital immunodeficiency: case reports and review of the literature
Response to Reviewer 1
Comment:
1 Good attempt by the authors, however the study lacks novelty as systemic retinoids are a known option for the management of warts especially in the setting of immunodeficiency.
2 Nicely written article, however focus of the manuscript should be on mechanism of action of retinoids in warts in general, in immunodeficiency disorders like GV, EDV and any specific proposed mechanism of action in IL-7 deficiency cases as in the present 2 cases.
5 Retinoids are given not only for the improvement of warts, but they also have roles as immunomodulators and prevention of skin cancers in conditions like EDV. Authors should mention this as well along with appropriate references.
Thank you so much for your valuable suggestions. There are no reports on the relationships between IL-7 and systemic retinoids in the setting of immunodeficiency disorders except for our cases (present report and Kosumi H et al Clin Infect Dis 2020). We have additionally mentioned the roles of retinoids as immunomodulators in the revised manuscript as suggested.
3 It would be good to mention the percentage improvement noted by care giver in the lesions. Also, mention the patient satisfaction with treatment, especially if it improved their quality of life.
In the case 1, we mentioned about the reduction in pain (line 66) and we have included her QOL improvement in the revised manuscript.
In the case 2, etretinate dramatically improved the lesions and flattened about 90% of them. Although the tapering of the etretinate caused the recurrence of the lesions, motor impairment and the pain during physical activities were markedly reduced to the extent that the lesions were controlled. We have included this information in the revised manuscript.
4 In table 1, mention the duration of therapy as well.
We have revised table 1 as suggested.
6 Since long term retinoids are generally needed for the management of warts in GV, a note on how to monitor therapy (like what investigations to be done and how frequently) and side effects of long term therapy should be mentioned for the benefit of readers.
We have performed serologic testing on the patients one month after the initiation of the treatment and repeated it every two to three months during the treatment. The serologic tests monitored liver function, kidney function and triglyceride metabolism, which can be affected by retinoids. We have added this information to the text.
7 Mention the success rate of improvement with retinoids for treatment of warts in general and in immunodeficiency disorders based on the available literature.
Although the efficacy of retinoids for the treatment of warts in general situation has been reported and evaluated (Thiele B et al, Z Hautkr, 1985, Gelmetti C et al, Pediatr Dermatol, 1987), it has not been reported ‘the rate of improvement’ of retinoids, possibly because it is difficult to evaluate the effectiveness due to its characteristics such as (i) high tendency of recurrence, (ii) difficulty in distinguishing recurrence from re-infection, (iii) combination with other therapeutics in the majority of cases. According to our review, it can be said that at least 11 out of 12 cases (91.7%) responded to the therapy, and we have added this information to the revised manuscript.
